# Epidemiology of Influenza-like Illness and Respiratory Viral Etiology in Adult Patients in Taiyuan City, Shanxi Province, China between 2018 and 2019

**DOI:** 10.3390/v15112176

**Published:** 2023-10-30

**Authors:** Zhao Jia, Puna Xue, Ruihong Gao, Rui Wang, Lifeng Zhao, Zhihong Zuo, Li Gao, Rui Han, Hong Yao, Jiane Guo, Jihong Xu, Zhen Zhu, Jitao Wang

**Affiliations:** 1School of Public Health, Shanxi Medical University, 56 Xinjian South Road, Taiyuan 030001, China; jz130966@163.com (Z.J.); xuepuna0108@163.com (P.X.); grhbb@foxmail.com (R.G.); yyao1976@163.com (H.Y.); jianeg1@163.com (J.G.); 2Taiyuan Center for Disease Control and Prevention, No. 22, Huazhang West Street, Xiaodian District, Taiyuan 030032, China; wangrui871020@163.com (R.W.); zhao929@163.com (L.Z.); zzh163@126.com (Z.Z.); gaolibiology@163.com (L.G.); 18734588040@163.com (R.H.); zyzd919@163.com (J.X.); 3NHC Key Laboratory of Medical Virology and Viral Diseases, National Institute for Viral Disease Control and Prevention, Chinese Center for Disease Control and Prevention, Beijing 102206, China

**Keywords:** influenza-like illness (ILI), epidemiology, influenza virus, respiratory viral pathogens, multiplex real-time polymerase chain reaction

## Abstract

To determine the epidemiological status of influenza and understand the distribution of common respiratory viruses in adult patients with influenza-like illness (ILI) cases in Taiyuan City, Shanxi Province, China, epidemiological data between 2018 and 2019 were retrieved from the China Influenza Surveillance Information System, and two sentinel ILI surveillance hospitals were selected for sample collection. All specimens were screened for influenza virus (IFV) and the other 14 common respiratory viruses using real-time polymerase chain reaction. The results of the 2-year ILI surveillance showed that 26,205 (1.37%) of the 1,907,869 outpatients and emergency patients presented with ILI, with an average annual incidence of 297.75 per 100,000 individuals, and ILI cases were predominant in children <15 years (21,348 patients, 81.47%). Of the 2713 specimens collected from adult patients with ILI, the overall detection rate of respiratory viruses was 20.13%, with IFV being the most frequently detected (11.79%) and at a relatively lower rate than other respiratory viruses. Further subtype analysis indicated an alternating or mixed prevalence of H1N1 (2009), H3N2, Victoria, and Yamagata subtypes. This study provides a baseline epidemiological characterization of ILI and highlights the need for a nationwide detection and surveillance system for multiple respiratory pathogens.

## 1. Introduction

Influenza-like illnesses (ILIs), a subset of acute respiratory infections (ARIs), account for approximately 62% of ARIs [1]. ILI can result in significant morbidity and mortality [2] and impose a substantial socioeconomic burden on families and society, particularly the elderly [3]. Many pathogens cause ARIs, including bacteria, viruses, mycoplasmas, chlamydia, and fungi [4]. Of these, viral pathogens are the most important causes of ILIs [5], and the most common is the influenza virus (IFV) [6]. Others include human respiratory syncytial virus (HRSV), human rhinovirus (HRV), human parainfluenza viruses (HPIV)1–4, human metapneumovirus (HMPV), human coronaviruses (HCoV) (subtype: NL63, 229E, OC43, and HKU1), human enterovirus (HEV), human adenovirus (HAdV), and human bocavirus (HBoV), which were also frequently identified in patients with either ARI or pneumonia, especially in children under 5 years of age [7,8].

Influenza is an acute respiratory infectious disease caused by IFV, characterized by sudden fever, cough, headache, muscle and joint pain, severe malaise, sore throat, and runny nose [9]. Influenza is often associated with outbreaks and epidemics worldwide [10]. However, since ARI caused by various viral pathogens exhibits clinical symptoms similar to those of influenza [11], clinical diagnosis is usually unreliable, and etiological diagnosis largely relies on laboratory testing. Therefore, during influenza surveillance, ARI with influenza-like manifestations is often referred to as ILI [11].

In China, all sentinel hospitals report ILI cases to the Chinese National Influenza Center every week, and the data are recorded by the “China Influenza Surveillance Information System”. Shanxi Province is located in Northern China, with Taiyuan as the capital city in the center of the province. Taiyuan City established an ILI surveillance system in 2009, which included two sentinel surveillance hospitals. Monitoring the prevalence of ILI cases is an indicator used to understand the epidemiological status of influenza in the population. In this study, the epidemiology of ILI in Taiyuan City, Shanxi Province between 2018 and 2019 was analyzed, and a study of the spectrum of respiratory viruses in adult patients with ILI was conducted. This study aimed to determine the epidemiological features of IFV and its subtypes, understand the distribution of other viral pathogens in patients with ILI, and highlight the importance of preventing and controlling local respiratory viral diseases.

## 2. Materials and Methods

### 2.1. ILI Definition

An ILI case refers to one in which a patient presents with a fever (≥38 °C), accompanied by coughing or a sore throat, and lacking other laboratory diagnostic evidence [8].

### 2.2. Data Collection

To analyze the epidemiological characteristics of ILI, data on ILI cases in Taiyuan City were obtained from the China Influenza Surveillance Information System, which included the number of outpatient and emergency cases, as well as case-based information, including the age of the cases and their time of onset. Demographic data were obtained from the Information System for All Health Security database. Overall incidence was defined as the total number of ILI cases divided by the average population size during the study period. The incidence rate of each age group was defined as the total number of ILI cases in each group divided by the average population size of the corresponding age group during the study period. The proportion of ILI cases was defined as the total number of ILI cases divided by the total number of outpatient and emergency department visits.

### 2.3. Specimen Collection

To investigate the etiology of respiratory viruses in patients with ILI in Taiyuan City, two sentinel ILI surveillance hospitals: the Taiyuan Central Hospital of Shanxi Medical University and the First Clinical Hospital of Shanxi Medical University, were selected for specimen collection from adult patients with ILI (≥18 years).

Based on the National Influenza Surveillance Technical Guidelines (2017 Edition), each sentinel hospital in Shanxi Province is required to collect 20 specimens per month from April to September and 10–40 specimens per week from October to March of the following year. Based on this principle, 2713 throat swab specimens from adult patients were collected between 2018 and 2019. All the specimens were collected within 3 days of onset, preserved in viral transport media, and sent to the Taiyuan Center for Disease Control and Prevention (Taiyuan CDC). All the specimens were transported at 2–8 °C and stored at −80 °C. Each specimen was assigned a unique laboratory code and recorded in the Influenza Surveillance Information Database at the Taiyuan CDC.

### 2.4. Nucleic acid Extraction

Viral RNA/DNA was extracted from specimens using a MagMAX™-96 Viral RNA/DNA Isolation Kit (Thermo Fisher Scientific, Foster City, CA, USA) according to the manufacturer’s instructions. RNA/DNA was eluted with 50 µL of elution buffer and used directly or stored at −80 °C.

### 2.5. IFV Identification

IFV typing was conducted using the IFVA and IFVB Real-Time RT-PCR Kit (Jiangsu BioPerfectus Technologies Co., Ltd., Taizhou, China). Subtype identification of IFVA- or IFVB-positive specimens was performed using commercially available real-time polymerase chain reaction (RT-PCR) kits (Nucleic Acid Detection Kit for H1N1 (2009)/H3N2 or Victoria/Yamagata, Jiangsu BioPerfectus Technologies Co., Ltd., Taizhou, China). The reaction solutions were prepared, cycling parameters were set in a CFX96 real-time thermal cycler (Bio-Rad, Hercules, CA, USA), and results were determined according to the manufacturer’s instructions.

### 2.6. Multiplex Real-Time RT-PCR/PCR Identification for Other Respiratory Viral Pathogens

All specimens were further screened for other respiratory viruses using multiplex real-time RT-PCR/PCR via the AgPath-ID™ One-Step RT-PCR kit (Thermo Fisher Scientific, Foster City, CA, USA). According to previously reported assays [12,13,14,15,16,17,18,19], each specimen was amplified in six parallel reactions containing eight viruses and four subtypes of HCoV and HPIV. Each reaction contained primers and probes specific for one to four targets (reaction A: HKU1, 229E, OC43; reaction B: HPIV1, HEV, HMPV, HRSV; reaction C: HPIV2, HAdV, HBoV; reaction D: NL63; reaction E: HRV; reaction F: HPIV3, HPIV4) (Table 1). All primers and probes were synthesized by Thermo Fisher Scientific, Inc. (Shanghai, China) (Table 1).

### 2.7. Statistical Analysis

Microsoft Office Excel 2019 was used to organize the data and draw charts. Statistical analysis was performed using SPSS 26.0 (SPSS Inc., Chicago, IL, USA), and chi-square tests were performed on the demographic categories of patients, with statistical significance set at *p* < 0.05.

## 3. Results

### 3.1. Epidemiological Feature

Of the 1,907,869 patients who visited the outpatient and emergency departments of two sentinel ILI surveillance hospitals in Taiyuan City between 2018 and 2019, 26,205 (1.37%) had ILI, with an average annual incidence of 297.75 per 100,000 individuals (2018: 12,453 patients, 284.33/100,000; 2019: 13,752 patients, 311.03/100,000). The reported ILI cases in Taiyuan City of Shanxi Province were predominantly in children <5 years (12,028 patients, 3055.84/100,000), followed by children aged 5–14 years (9320 patients, 1259.26/100,000) and adolescents and young adults aged 15–24 years (2861 patients, 57.62/100,000), and the elderly ≥60 years had the lowest incidence (577 patients, 45.00/100,000). The chi-square results of pairwise comparisons showed statistically significant differences in the incidence rates between age groups ($x^{2}$ = 138,186.977, *p* < 0.01) (Table 2).

### 3.2. Seasonal Characteristics of ILI

Between 2018 and 2019, the number of cases remained low between April and September (598–753 cases), after which it increased from October, and the epidemic peak (>1500 cases) occurred between January and March of the following year. This trend was also observed in early 2018 and late 2019 (Figure 1).

### 3.3. Overall Detection of Respiratory Viruses in Adult Patients with ILI

Between 2018 and 2019, we collected 2713 throat swab specimens from adult patients with ILI and identified 18 respiratory viruses (IFV: H1N1 (2009), H3N2, Yamagata, Victoria; HCoV: NL63, HKU1, 229E, OC43; HRSV; HRV; HPIV1-4; HMPV; HEV; HAdV; and HBoV) using multiplex real-time RT-PCR, showing that at least one respiratory virus was detected in 546 specimens (20.13%). Among them, the detection rate of IFV was dominant (11.79%), with H1N1 (2009) having the highest rate among all subtypes (7.04%), and the overall detection rates for HAdV (2.17%) and HRV (1.99%) were the highest. Nevertheless, the detection of several respiratory viruses varied between 2018 and 2019. For example, the rate of detection of five viruses (HPIV, HMPV, HEV, HCoV, and HBoV) in 2018 was 1.30–2.86%, whereas it was considerably lower in 2019 (0.08–0.88%). In addition, the HRSV detection rate was low in both years (0.68% and 0.48%) (Figure 2).

Of the 546 positive cases, 61 (61/546, 11.17%) were co-infected with other respiratory viruses, including 48 and 13 cases of dual and triple infections, respectively. Dual infections were mostly found in HRV and HEV (11/61, 18.03%), and HAdV and HBoV (9/61, 14.75%); triple infections were mainly observed in HAdV, HBoV, and HPIV2 (9/61, 14.75%).

### 3.4. Detection of IFV Subtypes Changed over Time

Of the 2713 specimens collected, 320 (11.80%) were identified as IFV-positive, of which 248 (9.14%) were IFVA and 72 (2.65%) were IFVB. Further subtyping showed that the detection rates of H1N1 (2009) and H3N2 were 7.04% and 2.10%, respectively, followed by Yamagata (1.77%) and Victoria (0.88%).

The results of the annual analysis showed that the major subtypes of IFV detected in 2018 were H1N1 (2009) (7.91%) and Yamagata (3.21%), whereas in 2019, an increase was observed in the detection rates of H3N2 (3.69%), Victoria (1.60%), and H1N1 (2009) (6.01%).

### 3.5. Age Distribution of Patients with Viral Infections

The 2713 adult patients with ILI were divided into three age groups (18–24, 25–59, and ≥60 years). Analysis of the distribution of respiratory viruses among the different age groups showed that H1N1 (2009) had the highest detection rate (4.28–8.09%) in all three groups, with the rate in the 25–59 age group (8.09%) being significantly higher than that in the other age groups, and significant differences in age groups were observed in H1N1 (2009) ($x^{2}$ = 10.068, *p* < 0.01) (Figure 3).

The predominant viruses detected in patients with ILI in different age groups varied, except for H1N1 (2009). For example, in patients aged 18–24, the detection rate of H3N2 (2.85%) and HAdV (2.54%) was higher, followed by Yamagata (2.22%) and HRV (1.90%); among patients aged 25–59, HAdV (2.35%), HRV (2.04%), HEV (1.91%), and H3N2 (1.73%) were more frequently detected than other viruses, and the HEV detection rate in this group was significantly higher than that in other age groups, with significant differences in age groups ($x^{2}$ = 6.937, *p* < 0.05). In elderly patients (≥60 years of age), the main viruses were H3N2 (7.14%) and HBoV (3.03%), and the HBoV detection rate in this group was significantly higher than that in other age groups, with significant differences in age groups ($x^{2}$ = 8.840, *p* < 0.05). Additionally, co-infection was observed in all three age groups, with detection rates ranging from 2.16% to 3.01% (Table 3).

### 3.6. Monthly Distribution of Respiratory Viruses

Between 2018 and 2019, the overall detection rate of IFV was consistent with the trend in the distribution of ILI, which is shown in Figure 1. Of the subtypes detected, IFVA had the highest detection frequency, and two major H1N1 (2009) peaks were observed, including the period between January and March 2018 and between December 2018 and February 2019. In addition, Yamagata (January–March 2018), H3N2, and Victoria (December 2018–April 2019) were simultaneously detected during these two periods. Another H3N2 peak was observed in December 2019. These results indicate that the IFV subtypes may vary during different epidemic seasons (Figure 3). No clear seasonality was observed for other respiratory viruses; however, these viruses were detected at relatively high rates during the non-influenza season (Figure 3).

**Figure 3 viruses-15-02176-f003:**
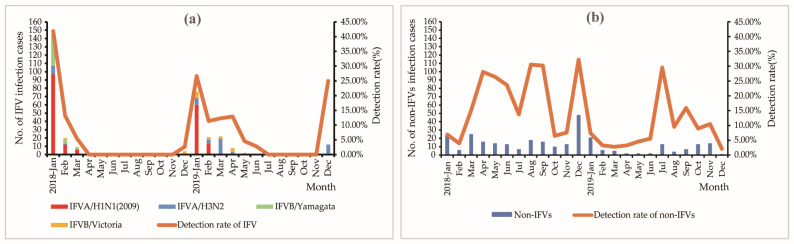
Monthly distribution of IFV (**a**) and non-IFVs (**b**) between 2018 and 2019.

## 4. Discussion

Based on data from the 2 year (2018–2019) surveillance of ILI, this study provided a preliminary landscape of the epidemiology of ILI in Taiyuan City, Shanxi Province, China. ILI cases were predominantly reported in children <15 years, and people ≥60 years were the least reported in Taiyuan City between 2018 and 2019. This susceptibility to ILI in younger age groups may be due to several reasons. First, children aged <5 years have a naive immune system and are more susceptible to infections. Second, adolescents aged 5–14 years congregate in schools, facilitating the spread of respiratory infections. Patients with ILI ≥60 years usually have underlying diseases, including hypertension, diabetes, or chronic bronchitis [20], and only seek medical attention when these primary diseases worsen; this group of patients might self-administer medications, which might account for the lower number of ILI cases among the elderly. Additionally, influenza epidemics tend to be seasonal and occur in winter and spring in Northern China [21], where dry and cold conditions are more suitable for IFV survival and transmission. As a Northern city, the influenza in Taiyuan City showed similar seasonal characteristics, suggesting that epidemiological surveillance and prevention and control efforts for ILI should be focused on winter and spring.

The overall detection rate of respiratory viruses in this study was 20.13%, which was lower than that reported in previous studies among adults in Northern China [22,23], and the detection rates may vary according to the geographical location and year of testing. IFV was the most frequently detected virus in patients with ILI, and between 2018 and 2019, influenza in Taiyuan City was characterized by alternating or mixed circulation of H1N1 (2009), H3N2, Victoria, and Yamagata, which was consistent with the findings of other studies [24]. This trend of alternating circulation of IFV subtypes may contribute to the ability of the virus to consistently cause influenza epidemics, which could be related to herd immunity against IFV subtypes. In addition, studies have shown that the COVID-19 pandemic reduced the circulation of seasonal influenza, while in recent seasons it has returned to pre-pandemic levels, and the seasonal distribution of influenza has been shown to be different than pre-pandemic cases [25]. A similar situation had also been observed in Taiyuan City in recent years. Data from the China Influenza Surveillance Information System showed that the detection rates of IFV in Taiyuan City were 5.5%, 3.6%, and 15.81%, respectively, during 2020–2022; meanwhile, the prevalent viral type had gradually shifted from the IFVA to IFVB, which became the predominant type during 2021–2022 (unpublished data). The transmissibility of influenza viruses during upcoming influenza epidemics would be affected by public health and social measures implemented globally since 2020 to mitigate the COVID-19 pandemic [26], making continuous influenza surveillance crucial.

The most effective way to prevent influenza is through vaccination [9], which can reduce the morbidity and mortality associated with influenza [27]. The best time for influenza vaccination is 1–2 months before the peak of the influenza epidemic, and the antibody produced after vaccination can last for 6–8 months [28]. The annual number of influenza cases in Taiyuan City gradually increased from October; therefore, September to October was the prime time for vaccination to avoid local large-scale epidemics of influenza. IFV undergoes rapid evolution by antigenic shift and drift [29]; therefore, developing a universal vaccine that covers a wider range of IFV strains is a long-term goal to guide countries to better respond to the next influenza season.

The World Health Organization Global Influenza Surveillance and Response System updates the composition of influenza vaccines twice a year [9]. During the northern hemisphere influenza season, the vaccine strains for IFVA recommended by WHO were the A/Michigan/45/2015 (H1N1) pdm09-like virus strain and the A/Hong Kong/4801/2014 (H3N2)-like virus strain during 2017–2018 [30], the A/Michigan/45/2015 (H1N1) pdm09-like virus strain and the A/Singapore/INFIMH-16-0019/2016 (H3N2)-like virus strain during 2018–2019 [31], and the A/Brisbane/02/2018 (H1N1) pdm09-like virus strain and the A/Kansas/14/2017 (H3N2)-like virus strain during 2019–2020 [32]. The data from the China Influenza Surveillance Information System showed that the strains of IFVA prevalent in Taiyuan City during 2018–2019 were generally matched with the WHO-recommended vaccine strains. In 2018 and 2019, the highest influenza vaccination rates in Taiyuan City were among children under 5 years of age (11.06% and 15.17%), followed by elderly people ≥60 years of age (1.52% and 2.30%), with lower rates in the remaining age groups (<1%) (unpublished data). Lower vaccination rates may lead to the accumulation of a large number of susceptible people, thus increasing the risk of an influenza pandemic. Therefore, the government departments should step up publicity efforts and encourage people to get vaccinated to avoid the occurrence of a regional pandemic of influenza.

Moreover, IFV and all eight other common respiratory viruses (HCoV, HRSV, HRV, HPIV, HMPV, HEV, HAdV, and HBoV) were detected, confirming that multiple respiratory viruses co-circulated in Taiyuan City. However, since these viruses had a significant disease burden in infants and young children, their lower detection rate in adult cases (maximum 2.17%) compared to IFV (11.80%) was reasonable in our study. Furthermore, the aforementioned respiratory viruses were detected primarily during the non-influenza epidemic season. Transient immune-mediated interference can cause common cold infections to diminish during the peak activity of a seasonal virus [33], which may explain why these viruses are less frequent during the influenza seasons. In addition, the lower prevalence of non-IFVs might increase random sampling errors and thus have a greater impact on the detection results.

Among the non-IFV respiratory viruses, the detection rates of HAdV (2.17%) and HRV (1.99%) in adult patients with ILI in Taiyuan City were relatively high. HRV and HAdV cause severe clinical manifestations and adverse consequences in adults. HRV is usually associated with exacerbations of asthma and chronic obstructive pulmonary disease [34], while HAdV is primarily related to community-acquired pneumonia; for example, HAdV infection has a longer duration of fever and higher rates of invasive ventilation and extracorporeal membrane oxygenation use [35]. Additionally, the study’s findings showed that HEV and HBoV infections were primarily discovered in young adults (18–24 years old) and elderly individuals (≥60 years old), respectively. These findings have also been reported in studies from other Chinese provinces and abroad [36,37,38,39,40,41,42]. Nevertheless, additional surveillance data are required for further confirmation, as the epidemiological profile of these viruses in adults remains incompletely understood.

The severity and disease burden of non-IFVs in adults remain uncertain, and there are still no effective specific vaccines or antiviral drugs for most non-IFV respiratory viruses, with supportive therapy being the mainstay of post-infection care [43,44]. Between November and December 2022, North America and several European countries experienced an unprecedented epidemic outbreak featuring the simultaneous circulation of SARS-CoV-2, IFV, and HRSV in children, which overloaded already constrained healthcare systems. To avoid this, establishing and conducting nationwide multiple respiratory pathogen detection and monitoring is required to understand the exact infection status and epidemiological situation of common respiratory viruses in adults and provide a scientific basis for developing preventive and control measures related to respiratory infections in adults. Moreover, co-infection was frequently observed between common viral respiratory pathogens and may be linked to more severe illnesses, particularly in immunocompromised populations [45]. Accordingly, although the proportion of co-infection cases was relatively low in this study, it should not be overlooked.

This study had some limitations: (1) Because of the limited surveillance years (2018 and 2019), the epidemiological pattern of ILI cases could not be fully understood, which might lead to bias in the analysis. (2) Only specimens from adult patients with ILI were detected in this study. Thus, the distribution of respiratory viruses across the entire population could not be obtained. (3) The types of respiratory viruses were not further identified. Thus, the relationship between viral type and ILI cases lacked in-depth analysis. (4) This study only analyzed common respiratory viruses and lacked information on other etiological agents.

## 5. Conclusions

This study analyzed the epidemiological characterization of ILI and common respiratory viral etiologies in adult patients with ILI in Taiyuan City, Shanxi Province, China, for the first time, providing baseline data for continuous monitoring. However, long-term laboratory-based surveillance should be conducted to better understand the etiology of ILI and to further reveal the epidemic features of respiratory viruses causing ILI in Taiyuan City, thus laying the etiological foundation for preventing and treating respiratory viruses in Taiyuan City.

## Figures and Tables

**Figure 1 viruses-15-02176-f001:**
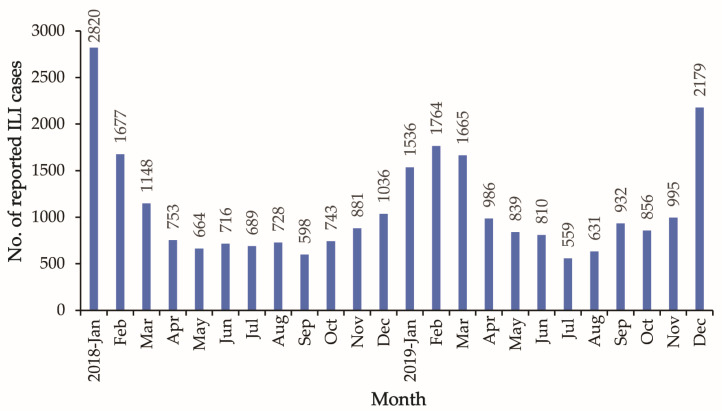
The monthly distribution of reported ILI cases between 2018 and 2019.

**Figure 2 viruses-15-02176-f002:**
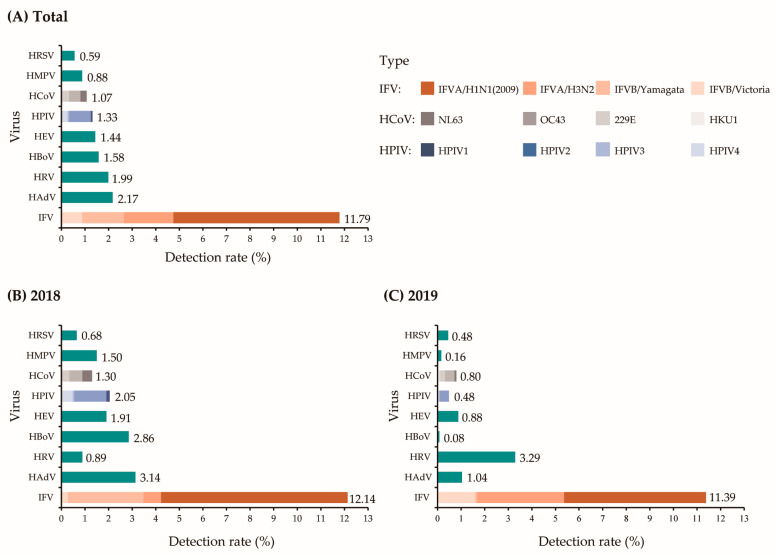
The detection rate of nine respiratory viruses between 2018 and 2019. (**A**) Summary data for 2018 and 2019; (**B**) data from 2018; (**C**) data from 2019.

**Table 1 viruses-15-02176-t001:** Sequences of primers and probes used in multiplex RT-PCR/PCR in the five reaction mixtures A–F.

Mixture	Pathogen	Primer/Probe	Sequences (5′-3′) *	Labels 5′/3′
A	HKU1	Forword primer	CACTTCTATTCCCTCCG TGTTTC	
		Reverse primer	TTAGAAGCAGACCTTCCTGAGCC	
		Probe	CGCCTGGTACGATTTTGCCTCAAGGCT	FAM/TAMRA
	229E	Forword primer	CAGTCAAATGGGCTGATGCA	
		Reverse primer	AAAGGGCTATAAAGAGAATAAGGTATTCT	
		Probe	CCCTGACGACCACGTTGTGGTTC	HEX/BHQ
	OC43	Forword primer	CGATGAGGCTATTCCGACTAGGT	
		Reverse primer	CCTTCCTGAGCCTTCAATATAGTAACC	
		Probe	TCCGCCTGGCACGGTACTCCCT	CY5/BHQ
B	HPIV1	Forword primer	GTTGTCAATGTCTTAATTCGTATCAATAATT	
		Reverse primer	GTAGCCTMCCTTCGGCACCTAA	
		Probe	TAGGCCAAAGATTGTTGTCGAGACTATTCCAA	FAM/TAMRA
	HEV	Forword primer	CATGGTGYGAAGAGTCTATTGAGCTA	
		Reverse primer	GGACACCCAAAGTAGTCGGTTC	
		Probe	CGGCCCCTGAATGCGGCTAATC	HEX/BHQ
	HMPV	Forword primer	ATGTCTCTTCAAGGGATTCACCT	
		Reverse primer	AMAGYGTTATTTCTTGTTGCAATGATGA	
		Probe	CATGCTATATTAAAAGAGTCTCARTAC	ROX/BHQ
	HRSV	Forword primer	GCAAATATGGAAACATACGTGAACA	
		Reverse primer	GCACCCATATTGTWAGTGATGCA	
		Probe	CTTCACGAAGGCTCCACATACACAGCWG	CY5/BHQ
C	HPIV2	Forword primer	GCATTTCCAATCTTCAGGACTATGA	
		Reverse primer	ACCTCCTGGTATAGCAGTGACTGAAC	
		Probe	CCATTTACCTAAGTGATGGAATCAATCGCAAA	FAM/TAMRA
	HAdV	Forword primer	GCCACGGTGGGGTTTCTAAACTT	
		Reverse primer	GCCCCAGTGGTCTTACATGCACATC	
		Probe	TGCACCAGACCCGGGCTCAGGTACTCCGA	HEX/BHQ
	HBoV	Forword primer	GGA AGA GAC ACT GGC AGA CAA	
		Reverse primer	GGG TGT TCC TGA TGA TAT GAG C	
		Probe	CTG CGG CTC CTG CTC CTG TGAT	ROX/BHQ
D	NL63	Forword primer	ACGTACTTCTATTATGAAGCATGATATTAA	
		Reverse primer	AGCAGATCTAATGTTATACTTAAAACTACG	
		Probe	ATTGCCAAGGCTCCTAAACGTACAGGTGTT	HEX/BHQ
E	HRV	Forword primer1	GGTGTGAAGAGCCSCRTGTGCT	
		Forword prime2	GGTGTGAAGACTCGCATGTGCT	
		Forword prime3	GGGTGYGAAGAGYCTANTGTGCT	
		Reverse primer3	GGACACCCAAAGTAGTYGGTYC	
		Probe	CCGGCCCTGAATGYGGCTAAYC	CY5/BHQ
F	HPIV4	Forword primer	CCTGGAGTCCCATCAAAAGT	
		Reverse primer	GCATCTATACGAACACCTGCT	
		Probe	GCTGCCGTCTCAAAATTTGTTGATCAAGACAATACAATTGGCAGC	ROX/BHQ
	HPIV3	Forword primer	GGAGCATTGTGTCATCTGTC	
		Reverse primer	TAGTGTGTAATGCAGCTCGT	
		Probe	CGCGCTACCCAGTCATAACTTACTCAACAGCAACAGCGCG	CY5/BHQ

Note: * M = A + C; R = A + G; S = C + G; Y = C + T; N = A + C + G + T; W = A + T.

**Table 2 viruses-15-02176-t002:** Data of reported ILI cases during 2018–2019.

Items	2018	2019	Total
No. of outpatient and emergency cases	920,635	987,234	1,907,869
No. of ILI reported cases (%)	12,453 (1.35)	13,752 (1.39)	26,205 (1.37)
Incidence rate (per 100,000)	284.33	311.03	297.75
Incidence by age groups (per 100,000)			
0–4	5839 (3161.20)	6189 (2962.60)	12,028 (3055.84)
5–14	4471 (1236.70)	4849 (1280.80)	9320 (1259.26)
15–24	603 (70.00)	816 (146.02)	1419 (99.91)
25–59	1247 (52.60)	1614 (62.25)	2861 (57.62)
≥60	293 (48.80)	284 (41.62)	577 (45.00)

**Table 3 viruses-15-02176-t003:** The distribution of viral pathogens according to age group in 2713 ILI cases.

Virus Detected	18–24 (n = 631)		25–59 (n = 1620)		≥60 (n = 462)		Total (n = 2713) No. (%)	*p* Value
	Total No. (%)	Coinfection No. (%)	Total No. (%)	Coinfection No. (%)	Total No. (%)	Coinfection No. (%)		
HRV	12 (1.90)	1 (0.16)	33 (2.04)	9 (0.56)	9 (1.95)	3 (0.65)	54 (1.99)	0.098
HAdV	16 (2.54)	6 (0.95)	38 (2.35)	16 (0.99)	5 (1.08)	4 (0.87)	59 (2.17)	0.202
HBoV	5 (0.79)	4 (0.63)	24 (1.48)	15 (0.93)	14 (3.03)	6 (1.30)	43 (1.58)	0.012
HEV	6 (0.95)	1 (0.16)	31 (1.91)	19 (1.17)	2 (0.43)	1 (0.22)	39 (1.44)	0.031
HMPV	6 (0.95)	0	14 (0.86)	5 (0.31)	4 (0.87)	0	24 (0.88)	0.98
HRSV	3 (0.48)	3 (0.48)	9 (0.56)	4 (0.25)	4 (0.87)	1 (0.22)	16 (0.59)	0.679
*IFV*	64 (10.14)	*0*	202 (12.47)	6 (0.37)	54 (11.69)	2 (0.43)	320 (11.80)	0.306
H1N1 (2009)	27 (4.28)	0	131 (8.09)	4 (0.25)	33 (7.14)	0	191 (7.04)	0.007
H3N2	18 (2.85)	0	28 (1.73)	1 (0.06)	11 (2.38)	0	57 (2.10)	0.223
Yamagata	14 (2.22)	0	25 (1.54)	1 (0.06)	9 (1.95)	2 (0.43)	48 (1.77)	0.523
Victoria	5 (0.79)	0	18 (1.11)	0	1 (0.22)	0	24 (0.88)	0.186
*HPIV*	4 (0.63)	4 (0.63)	24 (1.48)	13 (0.80)	2 (0.43)	2 (0.43)	36 (1.33)	0.071
HPIV1	0	0	2 (0.12)	1 (0.06)	0	0	2 (0.07)	
HPIV2	6 (0.95)	4 (0.63)	17 (1.05)	11 (0.68)	2 (0.43)	2 (0.43)	25 (0.92)	
HPIV3	0	0	1 (0.06)	0	1 (0.22)	0	2 (0.07)	
HPIV4	0	0	7 (0.43)	1 (0.06)	0	0	7 (0.26)	
*HCoV*	5 (0.79)	0	16 (0.99)	5 (0.31)	8 (1.73)	5 (1.08)	29 (1.07)	0.290
NL63	1 (0.16)	0	5 (0.31)	1 (0.06)	1 (0.22)	0	7 (0.26)	
OC43	2 (0.32)	0	7 (0.43)	2 (0.12)	4 (0.87)	3 (0.65)	13 (0.48)	
229E	2 (0.32)	0	3 (0.19)	2 (0.12)	3 (0.65)	2 (0.43)	8 (0.29)	
HKU1	0	0	1 (0.06)	0	0	0	1 (0.04)	
Total	119 (18.86)	15 (3.01)	337 (20.86)	35 (2.16)	90 (19.48)	11 (2.38)	546 (20.16)	

## Data Availability

The datasets used and/or analyzed in this study are available from the corresponding author upon reasonable request.

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
