# Peer review of "Epidemiology of Influenza-like Illness and Respiratory Viral Etiology in Adult Patients in Taiyuan City, Shanxi Province, China between 2018 and 2019"

_viruses, 2023, doi:10.3390/v15112176_

Round 1

Reviewer 1 Report

Comments and Suggestions for Authors

The authors tested thousands of clinical samples isolated from adult to investigate the pathogens which caused influenza-like illness in 2018 and 2019 in Taiyuan. There were totally 18 common flu-like viruses were screened by using the real-time PCR or RT-PCR. The data is informative. There are several comments needs to be addressed.

1.         It is should be mentioned that which circulating strains and which vaccine strains for the influenza A virus were used during 2018-2019, as well as the ratio and the age of the population get vaccinated in Taiyuan if possible. Then the authors can further discuss the prevention procedures based on the influenza A virus data.

2.         Are there any differences in the treatment between the different flu-like virus infection?

3.         The statement “epidemic” cannot be used for seasonal flu.

4.         Figure 2b and 2c, “Detction” should be “Detection”.

5.         Figure 3a could be further modified.

Comments on the Quality of English Language

The writing needs to be further polished.

Author Response

Dear Reviewer:

We are very grateful to your comments for our manuscript entitled “Epidemiology of influenza-like illness and respiratory viral etiology in adult patients in Taiyuan City, Shanxi Province, China between 2018 and 2019” (Manuscript ID: viruses-2655583). Those comments are all valuable and very helpful in revising and improving our paper. We have carefully read your comments and revised them.

Responds to the reviewer’s comments:

The authors tested thousands of clinical samples isolated from adult to investigate the pathogens which caused influenza-like illness in 2018 and 2019 in Taiyuan. There were totally 18 common flu-like viruses were screened by using the real-time PCR or RT-PCR. The data is informative. There are several comments needs to be addressed.

Point 1: It is should be mentioned that which circulating strains and which vaccine strains for the influenza A virus were used during 2018-2019, as well as the ratio and the age of the population get vaccinated in Taiyuan if possible. Then the authors can further discuss the prevention procedures based on the influenza A virus data.

------ Response to reviewer 1 comment No. 1: Thanks for your comments. Relevant information has been added to the Discussion part (Page 10, Line 242-257). The World Health Organization Global Influenza Surveillance and Response System updates the composition of influenza vaccines twice a year. During the northern hemisphere influenza season, the vaccine strains for IFVA recommended by WHO were A/Michigan/45/2015(H1N1) pdm09-like virus strain and A/Hong Kong/ 4801/ 2014(H3N2)-like virus strain during 2017-2018, A/Michigan/45/2015 (H1N1) pdm09-like virus strain and A/Singapore/ INFIMH-16-0019/2016 (H3N2)-like virus strain during 2018-2019, and A/Brisbane/02/2018 (H1N1) pdm09-like virus strain and A/Kansas/14/2017 (H3N2)-like virus strain during 2019-2020. The data from the China Influenza Surveillance Information System showed that the strains of IFVA prevalent in Taiyuan city during 2018-2019 were generally matched with the WHO-recommended vaccine strains. In 2018 and 2019, the highest influenza vaccination rates in Taiyuan city were among children under 5 years of age (11.06% and 15.17%), followed by elderly people ≥60 years of age (1.52% and 2.30%), with lower rates in the remaining age groups (<1%) (unpublished data). Lower vaccination rates may lead to the accumulation of a large number of susceptible people, thus increasing the risk of an influenza pandemic. Therefore, the government departments should step up publicity efforts and encourage people to get vaccinated to avoid the occurrence of a regional pandemic of influenza.

Point 2: Are there any differences in the treatment between the different flu-like virus infection?

------ Response to reviewer 1 comment No. 2: Thanks for your comments. There are still no effective specific vaccines and antiviral drugs for most of non-IFV respiratory viruses, with supportive therapy being the mainstay of post-infection care. It has been added in the Discussion part (Page 11, Line 276-278).

Point 3: The statement “epidemic” cannot be used for seasonal flu.

------ Response to reviewer 1 comment No. 3: Thank for your comments. Revised as suggested (Page 1 Line 15, Page 2 Line 57, 61, Page 10 Line 220,227,229).

Point 4: Figure 2b and 2c, “Detction” should be “Detection”.

------ Response to reviewer 1 comment No. 4: Thank for your comment. Revised as suggested (Figure 2b and 2c).

Point 5: Figure 3a could be further modified.

------ Response to reviewer 1 comment No. 5: Thank for your comment. Figure 3a has been modified.

Comments on the Quality of English Language: The writing needs to be further polished.

------ Response to reviewer 1 comment No. 6: The manuscript has been polished by the professional company before submission.

Reviewer 2 Report

Comments and Suggestions for Authors

Article summary and recommendation

The aim of this study was to determine the epidemic features of IFV and its subtypes, and understand the distribution of other viral pathogens in adult patients with ILI in Taiyuan City, Shanxi Province, China between 2018 and 2019. The authors have retrieved data from the China Influenza Surveillance Information system and two sentinel ILI surveillance hospitals were selected for sample collection. They have screened these samples for influenza virus, including subtyping for influenza A viruses and lineage typing for influenza B viruses by RT-PCR. Furthermore, they screen these samples for 14 other common respiratory viruses by RT-PCR. Their results show that the annual incidence of ILI is less than 1.5%, with pediatric patients showing the largest proportion of ILI cases. In adults, the overall detection of respiratory viruses in ILI case is low (~20%), and influenza viruses presented in more than half of these cases; the detection rate of influenza viruses followed the trend in distribution of ILI cases during the time period. The authors went further to characterize the distribution of seasonal influenza subtypes and lineages over the two year period. Their methodology is sound and they present the analysis of their results in a clear and concise manner. The article is well written. I agree with their conclusion that they highlight the need for influenza detection and surveillance systems and public health measures to curtail the spread of the disease. However, the study does lack some relevance to the field as it does not cover the most recent respiratory virus events/years. The COVID-19 pandemic had reduced the circulation of seasonal influenza, and in recent seasons it has returned to pre-pandemic levels, but the seasonal distribution of influenza has been shown to be different than pre-pandemic cases, doi: 10.1016/j.ijid.2022.08.002. Moreover, the transmissibility of influenza viruses during upcoming influenza epidemics would be affected by public health and social measures implemented globally since 2020 to mitigate the COVID-19 pandemic, https://doi.org/10.1016/S2214-109X(22)00358-8

The authors do not cover the above points in their study and data analysis or even during their discussion. This study would be greatly improved and be much more relevant to the field with ILI data and respiratory virus characterisation of ILI samples from recent years (2020-2023) in Taiyuan City.

Moreover, they claim that their study provides a baseline epidemiological characterization of ILI, but they do not show causative agents of the majority of ILI cases (the overall detection rate of respiratory viruses tested in this study in adult ILI cases is 20.13% - what are the major causes of ILI in adults?)

Specific points:

Line 61: Suggest re-wording text- “provide scientific data for preventing and controlling local respiratory viral diseases”. The authors don’t show scientific data for preventing or controlling respiratory virus diseases, they “provide scientific data” ‘to highlight the importance of’ “preventing and controlling respiratory virus diseases.”

Line 87: Re-word text to make clearer- the specimens were stored/transported at 2-8⁰C. Were the specimens preserved in viral transport media?

Table 2: The age 15-24 category overlaps both children and adult groups. Suggest to alter this category to 18-24 to make more consistent and easier to follow with subsequent sections focusing on adult ILI data, such as sections 3.3, 3.4 and 3.5

Line 131: not clear what is meant by incident rates here. In Table 2, the yearly incidence rate is expressed as number per 100,000 whereas for age groups, ILI incidence is expressed as a percentage (shown in brackets). Using the same method to express incidence rate would make the table clearer.

Section 3.4 : This section shows only a small proportion of adult ILI cases are caused by respiratory viruses. It would provide more insight into the disease epidemiology if the authors addressed in the discussion, the major causative agents of adult ILI cases.

Section 3.5: Lines 181-182 state that HEV case rate was significantly higher in 18-24 year olds than other age groups. Lines 183-185 state that the HBoV case rate was significantly higher in >60 year olds compared with other age groups. It would be beneficial to understanding the disease epidemiology if the authors discussed the reasons for these observed trends and provide comparisons with similar epidemiological studies.

Line 191: “consistent with the trend in the distribution of ILI,” ‘which is shown in Figure 1.’

Comments on the Quality of English Language

Well written and clear

Author Response

Responds to the reviewers:

Reviewer #2

The aim of this study was to determine the epidemic features of IFV and its subtypes, and understand the distribution of other viral pathogens in adult patients with ILI in Taiyuan City, Shanxi Province, China between 2018 and 2019. The authors have retrieved data from the China Influenza Surveillance Information system and two sentinel ILI surveillance hospitals were selected for sample collection. They have screened these samples for influenza virus, including subtyping for influenza A viruses and lineage typing for influenza B viruses by RT-PCR. Furthermore, they screen these samples for 14 other common respiratory viruses by RT-PCR. Their results show that the annual incidence of ILI is less than 1.5%, with pediatric patients showing the largest proportion of ILI cases. In adults, the overall detection of respiratory viruses in ILI case is low (~20%), and influenza viruses presented in more than half of these cases; the detection rate of influenza viruses followed the trend in distribution of ILI cases during the time period. The authors went further to characterize the distribution of seasonal influenza subtypes and lineages over the two year period. Their methodology is sound and they present the analysis of their results in a clear and concise manner. The article is well written. I agree with their conclusion that they highlight the need for influenza detection and surveillance systems and public health measures to curtail the spread of the disease. However, the study does lack some relevance to the field as it does not cover the most recent respiratory virus events/years. The COVID-19 pandemic had reduced the circulation of seasonal influenza, and in recent seasons it has returned to pre-pandemic levels, but the seasonal distribution of influenza has been shown to be different than pre-pandemic cases, doi: 10.1016/j.ijid.2022.08.002. Moreover, the transmissibility of influenza viruses during upcoming influenza epidemics would be affected by public health and social measures implemented globally since 2020 to mitigate the COVID-19 pandemic, https://doi.org/10.1016/S2214-109X(22)00358-8  

The authors do not cover the above points in their study and data analysis or even during their discussion. This study would be greatly improved and be much more relevant to the field with ILI data and respiratory virus characterisation of ILI samples from recent years (2020-2023) in Taiyuan City.

------Thanks for your comments. Relevant information has been added to the Discussion part (Page 11, Line 238-249).

In addition, studies have shown that the COVID-19 pandemic had reduced the circulation of seasonal influenza, while in recent seasons it has returned to pre-pandemic levels, and the seasonal distribution of influenza has been shown to be different than pre-pandemic cases. The similar situation had also been observed in Taiyuan city in recent years. Data from the China Influenza Surveillance Information System showed that the detection rates of IFV in Taiyuan city were 5.5%, 3.6%, and 15.81%, respectively, during 2020-2022; meanwhile, the prevalent viral type had gradually shifted from the IFVA to IFVB, which became the predominant type during 2021-2022 (unpublished data). The transmissibility of IFV during upcoming influenza epidemics would be affected by public health and social measures implemented globally since 2020 to mitigate the COVID-19 pandemic, highlighting continuous influenza surveillance crucial.

Moreover, they claim that their study provides a baseline epidemiological characterization of ILI, but they do not show causative agents of the majority of ILI cases (the overall detection rate of respiratory viruses tested in this study in adult ILI cases is 20.13% - what are the major causes of ILI in adults?)

------ Thanks for your comments. The overall detection rate of respiratory viruses tested in this study in adult ILI cases was 20.13%, and the detection rate of IFV was dominant (11.79%), with H1N1 (2009) having the highest rate among all subtypes (7.04%). It has been clarified in the Results (Page 6, Line 153-155) and Discussion part (Page11, Line 230-233).

Specific points

Line 61: Suggest re-wording text- “provide scientific data for preventing and controlling local respiratory viral diseases”. The authors don’t show scientific data for preventing or controlling respiratory virus diseases, they “provide scientific data” ‘to highlight the importance of’ “preventing and controlling respiratory virus diseases.”

------ Thanks for your comment. Revised as suggested (Page 2, Line 60-63). “This study aimed to determine the epidemiological features of IFV and its subtypes, understand the distribution of other viral pathogens in patients with ILI, and highlight the importance of preventing and controlling local respiratory viral diseases.”

Line 87: Re-word text to make clearer- the specimens were stored/transported at 2-8⁰C. Were the specimens preserved in viral transport media?

------ Thanks for your comment. Revised as suggested (Page 2, Line 89-92). “All the specimens were collected within 3 days of onset, preserved in viral transport media, and sent to the Taiyuan Center for Disease Control and Prevention (Taiyuan CDC). All the specimens were transported at 2-8⁰C and stored at -80⁰C.”

Table 2: The age 15-24 category overlaps both children and adult groups. Suggest to alter this category to 18-24 to make more consistent and easier to follow with subsequent sections focusing on adult ILI data, such as sections 3.3, 3.4 and 3.5

------ Thanks for your comment. The age distribution of the ILI cases was obtained directly from the China Influenza Surveillance Information System, in which age groups were categorized into five specific age groups: 0-4 years, 5-14 years, 15-24 years, 25-59 years, and ≥60 years. In this case, the age group of 15-24 years could not be changed to 18-24 years.

Line 131: not clear what is meant by incident rates here. In Table 2, the yearly incidence rate is expressed as number per 100,000 whereas for age groups, ILI incidence is expressed as a percentage (shown in brackets). Using the same method to express incidence rate would make the table clearer.

------ Thanks for your comment. Revised as suggested, the same method was used to express incidence rate (Page 2 Line 73-77, and Table 2). “Overall incidence was defined as the total number of ILI cases divided by the average population size during the study period. The incidence rate of each age group was defined as the total number of ILI cases in each group divided by the average population size of the corresponding age group during the study period.”

Section 3.4: This section shows only a small proportion of adult ILI cases are caused by respiratory viruses. It would provide more insight into the disease epidemiology if the authors addressed in the discussion, the major causative agents of adult ILI cases.

------ Thanks for your comment. It has been clarified in the manuscript “IFV was the most frequently detected virus in patients with ILI, and between 2018 and 2019, influenza in Taiyuan City was characterized by alternating or mixed circulation of H1N1(2009), H3N2, Victoria, and Yamagata, which was consistent with the findings of other studies”. (Page 11, Line 233-236)

Section 3.5: Lines 181-182 state that HEV case rate was significantly higher in 18-24 year olds than other age groups. Lines 183-185 state that the HBoV case rate was significantly higher in >60 year olds compared with other age groups. It would be beneficial to understanding the disease epidemiology if the authors discussed the reasons for these observed trends and provide comparisons with similar epidemiological studies.

------ Thanks for your comment. Relevant information has been added to the Discussion part (Page 12, Line 293-298). “Additionally, the study’s findings showed that HEV and HBoV infections were primarily discovered in young adults (18-24 years old) and elderly individuals (≥60 years old), respectively. These findings have also been reported in studies from other Chinese provinces and abroad. Nevertheless, additional surveillance data are required for further confirmation as the epidemiological profile of these viruses in adults remains incompletely understood.”

Line 191: “consistent with the trend in the distribution of ILI,” ‘which is shown in Figure 1.’

------ Thanks for your comment. Revised as suggested. The sentence has been changed to " Between 2018 and 2019, the overall detection rate of IFV was consistent with the trend in the distribution of ILI, which is shown in Figure 1". (Page 10, Line 199-200).

Reviewer 3 Report

Comments and Suggestions for Authors

The manuscript authored by Jia and colleagues presents findings on epidemic status of influenza and understand the distribution of common respiratory viruses in adult patients with influenza-like illness (ILI) cases in Taiyuan City, Shanxi Province, China, epidemiological data between 2018 and 2019.

This manuscript by Jia et al. has three major findings:

1) "The results of the 2-year ILI surveillance showed that 1.37% of the outpatients and emergency patients presented with ILI, with an average annual incidence of 297.75 per 100,000 individuals and ILI cases were predominant in children <15 years (21,348 patients, 81.47%);

2) Of the 2,713 specimens collected from adult patients with ILI, the overall detection rate of respiratory viruses was 20.13%, with IFV being the most frequently detected (11.79%) and relatively lower rate for other respiratory viruses;

3) The subtype analysis indicated an alternating or mixed prevalence of H1N1 (2009), H3N2, Victoria, and Yamagata subtypes."

From a scientific standpoint, the subject tackled by the authors is pertinent and encapsulates a contemporary health concern pertaining to influenza-like illness cases. Therefore, I didn't identify any issues with this manuscript. The approach, analysis, and presentation of results are well-done.

I have minor points that I see interesting to add to this manuscript:

1. Figures: The font of the figure captions should maintain the same style as the main text sections of the manuscript.

2. Figure 2 (B and C): Please considerer writing “Detection rate (%)” instead of “Detction rate (%)”. 

3. Table 3: It is not clear why the authors left some data contained in the table in bold. Please check this.

4. Discussion (Line 256): Please considerer writing “HAdV infection” instead of “HAdV-55 infection”.

Author Response

Dear Reviewer:

We are very grateful to your comments for our manuscript entitled “Epidemiology of influenza-like illness and respiratory viral etiology in adult patients in Taiyuan City, Shanxi Province, China between 2018 and 2019” (Manuscript ID: viruses-2655583). Those comments are all valuable and very helpful in revising and improving our paper.

Responds to the reviewer’s comments:

We have carefully read your comments and revised them. The manuscript authored by Jia and colleagues presents findings on epidemic status of influenza and understand the distribution of common respiratory viruses in adult patients with influenza-like illness (ILI) cases in Taiyuan City, Shanxi Province, China, epidemiological data between 2018 and 2019.

This manuscript by Jia et al. has three major findings:

1) "The results of the 2-year ILI surveillance showed that 1.37% of the outpatients and emergency patients presented with ILl, with an average annual incidence of 297.75 per 100,000 individuals and lLl cases were predominant in children <15 years (21.348 patients, 81.47%);

2) Of the 2,713 specimens collected from adult patients with ILl. the overall detection rate of respiratory viruses was 20.13%, with IFV being the most frequently detected (11.79%) and relatively lower rate for other respiratory viruses;

3) The subtype analysis indicated an alternating or mixed prevalence of HN1 (2009), H3N2, Victoria, and Yamagata subtypes.

From a scientific standpoint, the subject tackled by the authors is pertinent and encapsulates a contemporary health concern pertaining to influenza-like illness cases. Therefore, I didn't identify any issues with this manuscript. The approach, analysis, and presentation of results are well done.

I have minor points that I see interesting to add to this manuscript:

Point1: Figures: The font of the figure captions should maintain the same style as the main text sections of the manuscript.

------ Response to reviewer 2 comment No.1: Thanks for your comment. Revised as suggested. All the figure captions were modified to the same style as the main text section of the manuscript.

Point2: Figure 2 (B and C): Please considerer writing “Detection rate (%)” instead of “Detction rate (%)”.

------ Response to reviewer 2 comment No.2: Thanks for your comment. Revised as suggested.“Detction rate(%)” has been changed to “Detection rate (%)”.

Point3: Table 3: It is not clear why the authors left some data contained in the table in bold. Please check this

------ Response to reviewer 2 comment No.3: Thanks for your comment. Revised as suggested. The data contained in the tables have been removed from the bold form.

Point4: Discussion (Line 256): Please considerer writing “HAdV infection” instead of “HAdV-55 infection”.

------ Response to reviewer 2 comment No.4: Thanks for your comment. Revised as suggested. “HAdV-55 infection” was modified to “HAdV infection” (Page11, Line 274).

We appreciate for your warm work earnestly, and hope that the correction will meet with approval. Once again, thank you very much for your comments and suggestions.

Round 2

Reviewer 2 Report

Comments and Suggestions for Authors

thank you for addressing the reviewers comments